# Relaxing Equivariance Constraints with Non-stationary Continuous Filters

**Tycho F.A. van der Ouderaa**
Imperial College London
United Kingdom

**David W. Romero**
Vrije Universiteit Amsterdam
The Netherlands

**Mark van der Wilk**
Imperial College London
United Kingdom

## Abstract

Equivariances provide useful inductive biases in neural network modeling, with the translation equivariance of convolutional neural networks being a canonical example. Equivariances can be embedded in architectures through weight-sharing and place symmetry constraints on the functions a neural network can represent. The type of symmetry is typically fixed and has to be chosen in advance. Although some tasks are inherently equivariant, many tasks do not strictly follow such symmetries. In such cases, equivariance constraints can be overly restrictive. In this work, we propose a parameter-efficient relaxation of equivariance that can effectively interpolate between a (i) non-equivariant linear product, (ii) a strict-equivariant convolution, and (iii) a strictly-invariant mapping. The proposed parameterisation can be thought of as a building block to allow adjustable symmetry structure in neural networks. In addition, we demonstrate that the amount of equivariance can be learned from the training data using backpropagation. Gradient-based learning of equivariance achieves similar or improved performance compared to the best value found by cross-validation and outperforms baselines with partial or strict equivariance on CIFAR-10 and CIFAR-100 image classification tasks.

## 1 Introduction

Symmetric properties, such as equivariances and invariances, can be embedded into neural network architectures to provide inductive biases that leads to better data-efficiency and improved generalisation. Convolutional layers are known to provide translation equivariance in simple Euclidean spaces, and recent works have allowed various extensions to more complex groups and domains. However, symmetries are typically fixed, must be specified in advance, and can not be adjusted.

Symmetries embedded in network architectures enforce a hard constraint on the functions a neural network can represent. This can be an effective way to encode prior knowledge for problems that are inherently symmetric. However, hard symmetry constraints can become prohibitive if a problem does not strictly follow the symmetries. For example, convolutional layers can not encode potentially relevant absolute positional information, and '6's and a '9's become difficult to distinguish under rotation invariance. Relaxed symmetry constraints can mitigate such potential symmetry misspecification without losing the useful inductive biases that symmetries provide.

We propose to relax equivariance constraints by generalising the convolution operator with non-stationary filters that can also depend on absolute group elements. This results in a layer that can efficiently interpolate between (*i*) non-invariant linear products akin to a fully-connected layer, (*ii*) strict group equivariant convolutions, and (*iii*) strict group invariant mappings (Fig. 1).

The importance of this work is twofold. First of all, relaxable symmetry constraints can directly improve the performance in cases where strict symmetries are misspecified and result in an overly restrictive model class. Secondly, automatically learning symmetry structure from data is an interesting problem. Work in this field often focuses on invariances [34, 4, 33, 29, 12], which are easier to parameterise than equivariances. We show a way to effectively parameterise learnable equivariance constraints and demonstrate gradient-based learning of layer-by-layer equivariance constraints.

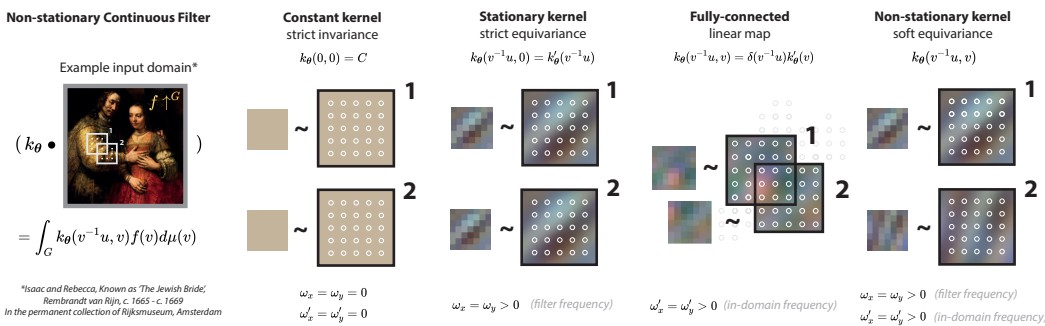

Figure 1: The non-stationary integral operator. Strict invariance, strict equivariance, non-equivariance, and relaxed equivariance are all special cases that depend on the frequency parameters $\boldsymbol{\omega}$ and $\boldsymbol{\omega'}$. The case of the translation group $G=\mathrm{T}(2)$ is illustrated: unlike the regular convolution with stationary kernels, the proposed non-stationary filter can depend on the position where it is applied.

## 2    Related Work

**Group equivariance.**    Equivariance constraints applied to layers provide a strong inductive bias that enforce transformations in the input to result in equivalent transformations in the output. For compact groups it can be shown that this constraint naturally leads to (group) convolutions [15]. Many works have allowed for equivariances to groups other than translation, including continuous roto-translations [38, 36, 16], discrete roto-translations [5, 3], scale [37], and permutations [39] and non-Euclidean domains, such as spheres [6], points clouds [10] and graphs [27].

**Symmetry misspecification and approximate equivariance.**    Although symmetry constraints can be very effective in machine learning, they can become prohibitive if data does not exactly follow the enforced symmetry. For instance, Liu et al. [17] raised an 'intriguing failing' of convolutional neural networks by showing that they can not encode absolute coordinate information. Adding explicit coordinate information was proposed as an ad-hoc solution to the problem, effectively breaking strict equivariance constraints. To achieve the same result, Finzi et al. [8] proposed to break equivariance of a convolutional layer by summing it with a non-equivariant fully-connected layer, requiring many additional parameters. Alternatively, Romero and Lohit [24] consider a sparse local support around the group identity element to break equivariance. This is effective for sparsely sampled subgroups, such as rotation, but becomes less practical for more densely sampled groups such as translation where the local support could lead to very sparse feature maps. Lastly, Wang et al. [35] propose a relaxed convolutional operator similar to our non-stationary approach, but require a low-rank factorisation which potentially limits the expressivity of the feature maps. The same work briefly discusses a mathematical description without such factorisation similar to our proposal which is deemed "too large a trainable weight space to be practical". This is true for discrete convolutional kernels. Instead we define our kernels as continuous kernels [26] parameterised with a finite number of parameters. Consequently, our weight space becomes tractable and does not pose a problem at all.

**Physics and filtering.**    Symmetries play a central role in physics. In [35] approximate equivariances are used to allow more robust models for dynamical systems. In Noether networks [1] conservation laws are inferred by learning symmetries from data. Our work could offer new insights in how symmetry constraints could be relaxed in such applications. In geophysics and seismology, there are some interesting parallels between non-stationary filtering methods and approximate equivariances in machine learning. In particular, [19] discusses generalisations of the convolution in the context of non-stationary filtering very similar to how we propose to relax equivariances in neural networks.

**Automatic symmetry discovery.**    Automatically inferring symmetry structure of neural network architectures from data is an interesting open problem. For instance, Lorraine et al. [18] learn data augmentations by differentiating the validation loss using the implicit function theorem and Zhou et al. [40] showed how symmetry structure could be learned through a meta-learning outer loop. In

[4] and [8] symmetries are selected using a training loss directly, using an additional regulariser. Symmetry discovery methods often focus on learning global invariances [34, 4, 29, 33, 12], which are easier to parameterise compared to layer-wise equivariances. We propose a way to continuously relax equivariances allowing efficient parameterisation of learnable equivariance for symmetry discovery methods. Inspired by the regularisation used in [4] to learn augmented inputs, we demonstrate that our parameterisation can be used for gradient-based learning of equivariance in each layer from data.

## 3 Background

### 3.1 Groups

A Lie group provides a natural way to describe continuous symmetry, as it forms a continuous manifold of which underlying elements are equipped with a group structure. Lie groups do not necessarily form a vector space. However, to every Lie group $G$ we can associate an underlying vector space called the Lie algebra $\mathfrak{g}$. The Lie algebra corresponds to the tangent space of the group at the identity, has the same dimensionality as the group, and captures the local structure of the group. Because the Lie algebra is a vector space, elements $\boldsymbol{a} \in \mathfrak{g}$ can be expanded in a basis in the Lie algebra $\boldsymbol{a} = \sum_{i=1}^{\dim(G)} \alpha_i A_i$ with coefficients $\boldsymbol{\alpha} \in \mathbb{R}^{\dim(G)}$. The exponential map $\exp : \mathfrak{g} \to G$ maps elements from the Lie algebra to the Lie group. We can also define a logarithm $\log : G \to \mathfrak{g}$ map from the Lie group to the Lie algebra. Such a choice always exists, but is not necessarily smooth or unique.

### 3.2 Equivariance and invariance

Equivariance is the property of a mapping such that transformations to the input result in equivalent transformations to the output. If changes are invertible, they can be described as the action of a group $G$ on some space $\mathcal{X}$. Formally, we say that a function $h : \mathcal{X} \to \mathcal{X}$ is *equivariant to the group $G$* if $h(g \cdot x) = g \cdot h(x)$ for all $g \in G, x \in \mathcal{X}$.

If the output of the function is completely independent to the action of the group $G$ on the input, we say that the function is invariant to the group $G$. Formally, a function $h : \mathcal{X} \to \mathcal{X}$ is *invariant to the group $G$* if $h(g \cdot x) = h(x)$ for all $g \in G, x \in \mathcal{X}$.

### 3.3 The group convolution

Let $f : G \to \mathbb{R}$ be an input signal and $k_{\boldsymbol{\theta}} : G \to \mathbb{R}$ be a convolutional kernel parameterised by $\boldsymbol{\theta}$. The group convolution is defined as:

$$h(u) = (k_{\boldsymbol{\theta}} \star f)(u) = \int_G k_{\boldsymbol{\theta}}(v^{-1}u)f(v)\mathrm{d}\mu(v), \tag{1}$$

where we integrate with respect to the Haar measure of the group $\mu$. Group convolutional structure is not just a sufficient, but also a necessary condition for equivariance to the action of a compact group [15]. The regular convolution for translation equivariance is the special case where the group is the translation group $G = \mathrm{T}(\mathrm{n}) \cong \mathbb{R}^n$, and the group action is given by addition, i.e., $v^{-1}u = u - v$.

The convolutional kernel $k_{\boldsymbol{\theta}}(v^{-1}u)$, which we will sometimes refer to as 'filter' for stylistic purposes, is called *stationary* because it can only depend on $u$ and $v$ through $v^{-1}u$. This invariance of the kernel is an important constraint and in fact a requirement for $h$ to be *equivariant*. Breaking the stationarity of the kernel allows us to relax the equivariant symmetry constraints, which is one of the central ideas of this paper.

### 3.4 Group lifting

The input and output signals of a neural network are typically not defined on the group $G$, but rather on an input and output space $\mathcal{X}$ and $\mathcal{Y}$. Using a group lifting and projection procedure [15, 7], we can define our model on the group and still apply it on the input and output space. We define a lifting operator $\uparrow^G$ to map input signals $f : \mathcal{X} \to \mathbb{R}$ to functions on the group $f \uparrow^G : G \to \mathbb{R}$. Similarly, we define a projection operator that maps function on the group $f' : G \to \mathbb{R}$ to functions on the output space $f' \downarrow_{\mathcal{Y}} : \mathcal{Y} \to \mathbb{R}$.

## 4 Method

### 4.1 Non-stationary Integral Operator

The group convolution operation of Eq. (1) is strictly equivariant due to the stationarity of the kernel. That is, the kernel $k$ only depends on relative group element $v^{-1}u$. For translation $G=\mathrm{T}(2)$, the kernel only depends on relative coordinates $(u-v)$ and not on the absolute position in the image. To relax equivariance constraints, we let the kernel also depend on the absolute group element $v$:

$$h(u) = (k_{\boldsymbol{\theta}} \bullet f)(u) = \int_G k_{\boldsymbol{\theta}}(v^{-1}u, v)f(v)\mathrm{d}\mu(v). \tag{2}$$

The main difference with the group convolution of Eq. (1) is that the kernel $k_{\boldsymbol{\theta}} : G \times G \to \mathbb{R}$ now has two input group elements. We will refer to the first input argument $v^{-1}u \in G$ as the *stationary component* and the second input argument $v \in G$ as the *non-stationary component*. In case of translation, this change lets the kernels depend also on the absolute coordinate of the input image on which it is applied. Consequently, $h(u)$ does no longer describe the regular convolution, but rather a non-stationary integral operator, for which the convolution remains a limiting case.

For the purpose of this work, there is no meaningful difference between expressing the non-stationary component in the input domain $k_{\boldsymbol{\theta}}(v^{-1}u, v)$ or the output domain $k_{\boldsymbol{\theta}}(v^{-1}u, u)$. For consistency, we will stick with the input domain throughout this work. One could also think of writing the kernel in an even more general form: $k_{\boldsymbol{\theta}}(u, v)$. This, as such, does not lead to a more expressive kernel in the sense that both forms can represent the same class of functions (for a proof, see App. C.1). Further, in this more general way of writing the non-stationary kernel it will become harder to formulate a parameterisation that allows for controllable symmetry constraints, so we do not consider it.

The non-stationary integral operator of Section 4.1 can represent interesting special cases through an independence of $k_{\boldsymbol{\theta}}$ with respect to the first and second input arguments.

**Remark 1** (Linear product / Fully-connected). *If the kernel equals an impulse response in the stationary component multiplied by some function $k'_{\boldsymbol{\theta}}$ in the non-stationary component $k_{\boldsymbol{\theta}}(v^{-1}u, v) = \delta(v^{-1}u)k'_{\boldsymbol{\theta}}(v)$, then the operator $h(u) = (k_{\boldsymbol{\theta}} \bullet f)(u)$ corresponds to a linear product with $k'_{\boldsymbol{\theta}}$.*

If we have a kernel $k_{\boldsymbol{\theta}}(v^{-1}u, v) = \delta(v^{-1}u)k'_{\boldsymbol{\theta}}(v)$, then we have that:

$$h_{\boldsymbol{\theta}}(u) = \int_G k_{\boldsymbol{\theta}}(v^{-1}u, v)f(v)\mathrm{d}\mu(v) = \int_G \delta(v^{-1}u)k'_{\boldsymbol{\theta}}(v)f(v)\mathrm{d}\mu(v) = k'_{\boldsymbol{\theta}}(u)f(u), \tag{3}$$

which is the linear product between $k'_{\boldsymbol{\theta}}(u)$ and input $f(u)$. This is akin to an element-wise product continuously defined on $G$. Optionally, this could be followed by an invariant layer to average pool activations. If the operation is performed over channels and we regard these as output features, this corresponds to a fully-connected layer in the discrete case $G=\mathbb{Z}^n$, with weights given by all $k'_{\boldsymbol{\theta}}$.

**Remark 2** (Group equivariance / Convolution). *If the kernel does not depend on the non-stationary component, then the operator $h(u) = (k_{\boldsymbol{\theta}} \bullet f)(u)$ is equivalent to a strict G-equivariant convolution.*

If the kernel only depends on the first stationary component, then $k_{\boldsymbol{\theta}}(v^{-1}u, v) = k'_{\boldsymbol{\theta}}(v^{-1}u)$, and:

$$h_{\boldsymbol{\theta}}(u) = \int_G k_{\boldsymbol{\theta}}(v^{-1}u, v)f(v)\mathrm{d}\mu(v) = \int_G k'_{\boldsymbol{\theta}}(v^{-1}u)f(v)\mathrm{d}\mu(v) = (k'_{\boldsymbol{\theta}} \star f)(u), \tag{4}$$

which equals the group convolution which is known to be strictly group equivariant.

**Remark 3** (Group Invariance / Pooling). *If the kernel is independent of both stationary and non-stationary components, the operator $h(u) = (k_{\boldsymbol{\theta}} \bullet f)(u)$ is strictly G-invariant.*

If the kernel does not depend on both inputs, it must be constant $k_{\boldsymbol{\theta}}(v^{-1}u, u) = C$, and we obtain:

$$h_{\boldsymbol{\theta}}(u) = \int_G k_{\boldsymbol{\theta}}(v^{-1}u, u)f(v)\mathrm{d}\mu(v) = C' \int_G f(v)\mathrm{d}\mu(v), \tag{5}$$

which is a scaled average 'global-pooling' over group actions and leads to strict group invariance. We obtain partial invariance or 'local-pooling' if the kernel is only non-zero constant in some local support of the stationary domain.

## 4.2 Parameterising the kernel

**Lie algebra basis.** In Eq. (2), we have proposed a non-stationary integral operator with a kernel $k_{\boldsymbol{\theta}} : G \times G \to \mathbb{R}$ that takes two input arguments: the stationary group element $v^{-1}u$ and a non-stationary group element $v$. We take the approach of parameterising the kernel $k_{\boldsymbol{\theta}}$ in terms of real vector spaces, through an explicit Lie algebra basis, which are practical to work with. To do so, we first define a logarithm function $\log$ and express the kernel $k = \hat{k}(\log g, \log h)$ in terms of a kernel in the Lie algebra $\hat{k} : \mathfrak{g} \times \mathfrak{g} \to \mathbb{R}$, which always form vector spaces. For rotation $G$=SO(2), the exponential map $\exp$ sending Lie algebra elements (on a line) to Lie group elements (on a circle) is not left invertible, but is right invertible, as a function. As the right inverse is not unique, we choose to use the principal $\log$ sending Lie group elements to Lie algebra elements closest to the identity. In general, such a choice for the logarithm always exists but is not necessarily unique or continuous. We choose a vector space basis $\{A_i\}_{i=1}^{\dim(G)}$ to express Lie algebra elements $\boldsymbol{a} \in \mathfrak{g}$ in terms of coefficients $\boldsymbol{\alpha} \in R^{\dim(G)}$ that relate to Lie group elements $g \in G$ by:

$$g = \exp(\boldsymbol{a}) = \exp \sum_{i=1}^{\dim(G)} \alpha_i A_i \in G, \qquad \boldsymbol{\alpha} \in \mathbb{R}^{\dim(G)} \qquad (6)$$

Through this construction, each group element $g$ corresponds through a choice of $\log$ to a unique Lie algebra element $\boldsymbol{a}$ which can be implemented as a real vector $\boldsymbol{\alpha}$.

**Functions in the Lie algebra.** We parameterised the non-stationary kernel $k_{\boldsymbol{\theta}} : G \times G \to \mathbb{R}$, taking two group elements as input, in the Lie algebra $\hat{k} : \mathfrak{g} \times \mathfrak{g} \to \mathbb{R}$ acting on a product space of two vector spaces. We refer to the first vector space related to the stationary component as the *filter space*, as it corresponds to space in which filters of conventional convolutions are defined. And refer to the second vector space of the non-stationary component as the *domain space*, because elements directly relate to absolute group elements in the input domain.

In Section 4.1 we showed that an independence of the kernel with respect to the first and second input argument correspond to strict equivariant and invariant symmetry constraints. We would, therefore, like to choose $\hat{k}$ such that we can control the dependence on the stationary and non-stationary components, as this will allow for explicit control over symmetry constraints. To do so, we parameterise $\hat{k}$ using a set of random Fourier features [23] with a frequency parameter $\omega$ for each dimension. The frequency parameter gives us control over the spectral properties of the function in filter space and domain space. But crucially, a frequency of $\omega$=0 results in a constant infinite lengthscale and therefore a function $\hat{k}$ that is independent to the associated input dimension. This mechanic allows us to select the special cases discussed in Section 4.1 and interpolate between them! Together, we have a frequency parameter $\boldsymbol{\omega} \in \mathbb{R}^{\dim(G)}$ that controls spectral properties in filter space and a frequency parameter $\boldsymbol{\omega}' \in \mathbb{R}^{\dim(G)}$ to control spectral properties in domain space. We propose a weight-space parameterisation that allows for explicit control over the symmetry properties of the layer through frequency parameters $\boldsymbol{\omega}$ and $\boldsymbol{\omega}'$.

**Weight-space implementation.** We choose to parameterise our kernel in a finite $D$-dimensional random Fourier features (RFF) [23, 31] basis $\boldsymbol{\gamma}_{\boldsymbol{\omega}} : \mathfrak{g} \to \mathbb{R}^{2D}$ to embed Lie algebra elements $\boldsymbol{a} \in \mathfrak{g}$ as real vectors using their coefficients $\boldsymbol{\alpha} \in R^{\dim(G)}$ where the frequency can be explicitly controlled by frequency parameter $\boldsymbol{\omega} \in \mathbb{R}^{\dim(G)}$. Values of $\boldsymbol{W} \in \mathbb{R}^{D \times \dim(G)}$ are randomly initialised and can be kept fixed. We parameterise the Lie algebra kernel $\hat{k} : \mathfrak{g} \times \mathfrak{g} \to \mathbb{R}$ with a neural network $\mathrm{NN}_{\boldsymbol{\theta}} : \mathbb{R}^{4D} \to \mathbb{R}$ taking concatenated Fourier features as input:

$$\boldsymbol{\gamma}_{\boldsymbol{\omega}}(\boldsymbol{a}) = \sqrt{\frac{1}{D}} \begin{bmatrix} \cos\left(2\pi(\boldsymbol{W}(\boldsymbol{\alpha} \odot \boldsymbol{\omega}))\right) \\ \sin\left(2\pi(\boldsymbol{W}(\boldsymbol{\alpha} \odot \boldsymbol{\omega}))\right) \end{bmatrix}, \qquad \hat{k}_{\boldsymbol{\theta}}(\boldsymbol{a}_{v^{-1}u}, \boldsymbol{a}_v) = \mathrm{NN}_{\boldsymbol{\theta}}\left(\begin{bmatrix} \boldsymbol{\gamma}_{\boldsymbol{\omega}}(\boldsymbol{a}_{v^{-1}u}) \\ \boldsymbol{\gamma}_{\boldsymbol{\omega}'}(\boldsymbol{a}_v) \end{bmatrix}\right). \qquad (7)$$

where $\boldsymbol{\gamma}_{\boldsymbol{\omega}}(\boldsymbol{a}_{v^{-1}u})$ are the Fourier features on the filter space and $\boldsymbol{\gamma}_{\boldsymbol{\omega}'}(\boldsymbol{a}_v)$ are the Fourier features on the domain space, respectively parameterised by $\boldsymbol{\omega}$ and $\boldsymbol{\omega}'$ frequency parameter vectors.

Directly representing the kernel with an MLP on $\boldsymbol{\alpha}$ without Fourier features will likely hamper performance as MLPs are known to suffer from 'spectral bias' [2, 22], making it difficult to encode high-frequency functions. Sinusoidal activations in random Fourier features [23, 31] and SIRENs [30] alleviate this issue and have been found suitable encodings of positional information in neural networks [32], including (place-coded) continuous kernels in neural networks [26, 25].

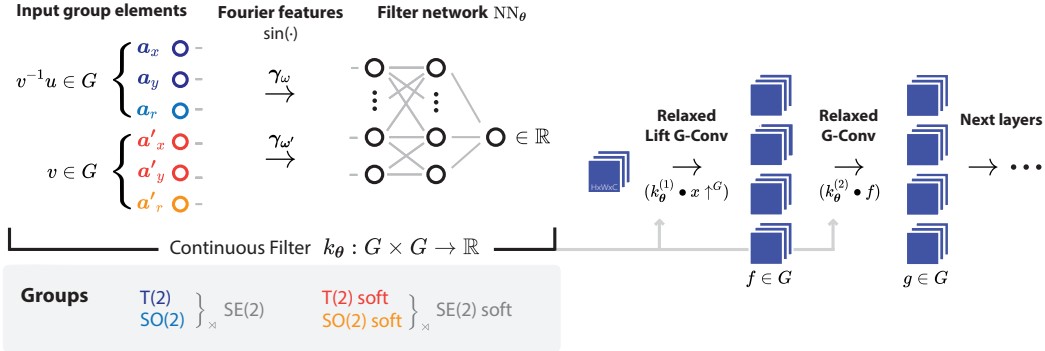

Figure 2: Parameterisation of continuous filter $k_{\boldsymbol{\theta}}$ in soft-SE$(2)$ equivariant model. Fourier features $\boldsymbol{\gamma_\omega}$ and $\boldsymbol{\gamma_{\omega'}}$ represent the stationary and non-stationary input components $v^{-1}u \in G$ and $v \in G$. Frequency parameters $\boldsymbol{\omega}$ and $\boldsymbol{\omega'}$ respectively control the spectra of the filter space and domain space, generalising the convolution to an operator with continuously adjustable symmetry constraints.

**Intuition behind frequency parameters.** The frequency parameters $\boldsymbol{\omega}$ and $\boldsymbol{\omega'}$ directly control the bases of the respective 'filter space' and 'domain space' bases. Specific values of $\boldsymbol{\omega}$ and $\boldsymbol{\omega'}$ correspond to a layer $h$ that exactly is or interpolates between special cases discussed in Section 4.1. Most notably, for $\boldsymbol{\omega'}{=}\mathbf{0}$ the layer becomes a strictly equivariant convolution and for both $\boldsymbol{\omega}{=}\boldsymbol{\omega'}{=}\mathbf{0}$ the layer performs invariant pooling. To understand why this is, notice that a feature with zero frequency parameter is constant $\boldsymbol{\gamma_0}$, by definition. For $\boldsymbol{\omega'}{=}\mathbf{0}$ we therefore have a constant feature for the non-stationary component $\boldsymbol{\gamma_{\omega'}}(\boldsymbol{a}_v){=}\boldsymbol{\gamma_0}$ and a kernel that is solely defined in filter space. If both $\boldsymbol{\omega}{=}\boldsymbol{\omega'}{=}\mathbf{0}$, both Fourier features become constant $\boldsymbol{\gamma_\omega}{=}\boldsymbol{\gamma_\omega}(\boldsymbol{a}_{v^{-1}u}){=}\boldsymbol{\gamma_0}$ and the entire kernel must therefore be constant, resulting in invariant pooling. Positive values for $\boldsymbol{\omega'}$ introduce non-stationarity that effectively relaxes the strict equivariance constraints of the convolution at $\boldsymbol{\omega'}{=}\mathbf{0}$. Lastly, note that individual scalars in $\boldsymbol{\omega'}$ correspond to particular subgroups and we can relax equivariance constraints of specific subgroups by letting $\boldsymbol{\omega'}$ be greater than zero in the associated dimension.

**Example for translation group** As example, we consider relaxing translation equivariance $G{=}\mathrm{T}(2)$. Since $\dim(\mathrm{T}(2)) = 2$, we have a two dimensional filter and domain spaces and therefore also two-dimensional filter space frequencies $\boldsymbol{\omega} = [\omega_x, \omega_y]^T \in \mathbb{R}^2$ and two domain space frequencies $\boldsymbol{\omega'} = [\omega'_x, \omega'_y]^T \in \mathbb{R}^2$. If $\boldsymbol{\omega'}{=}[0,0]^T$, the layer reduces to a translationally equivariant convolutional layer. In other words, for any value of $\boldsymbol{\omega}$ or weights $\boldsymbol{\theta}$ the kernel remains stationary (as we have $\boldsymbol{\omega'}{=}\mathbf{0}$). In this case the layer equals a convolution where the same filter is applied to all locations in the input. Higher values of $\boldsymbol{\omega}$, i.e. $\omega_x, \omega_y > 0$, correspond to more rapid changes in x- and y-direction in filter space, but the overall operation remains a convolution with a filter that is applied to all locaitons in the image. If, instead, we let the components of the domain space frequencies $\boldsymbol{\omega'}$ become non-zero, i.e. $\omega'_x, \omega'_y > 0$, then the kernel values can also depend on the absolute coordinate in the input image. Higher $\boldsymbol{\omega'}$ result in a more rapidly changing kernel relative to different absolute locations in the input image. This non-stationarity breaks equivariance constraints in a continuous way. For sufficiently high $\boldsymbol{\omega'}$, the function can become akin to a linear product that has enough variance to place an independent weight on each pixel location. In practice, this would also require a sufficiently flexible NN$_{\boldsymbol{\theta}}$. Different symmetry modes for translation $G{=}\mathrm{T}(2)$ are illustrated in Fig. 1.

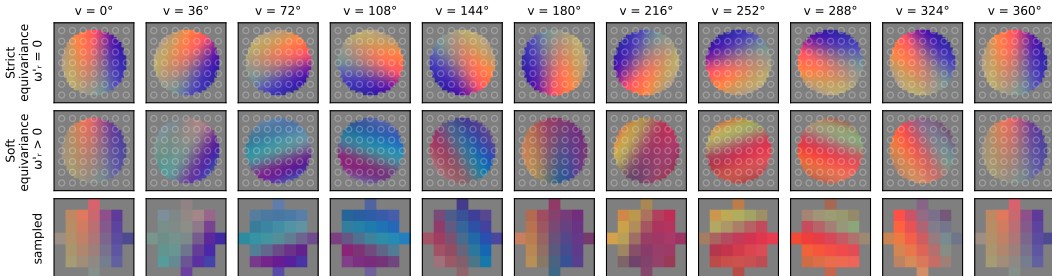

Figure 3: Equivariant and soft-equivariant SO$(2)$ filter banks. The top row shows a strict rotation equivariant filter bank for $\omega'_r{=}0$. The middle row shows a filter bank with positive non-stationary frequencies $\omega'_r{>}0$, without strict equivariance constraints. Sampled group elements at the bottom.

Table 1: Image classification test accuracies. Comparison between strict and relaxed equivariant models. Mean and standard error $\frac{\sigma}{\sqrt{3}}$ are reported over three different random seeds.

| Group | No. samples in SO(2) | T(2) | ⋊ SO(2) | Model | CIFAR-10 no augment | CIFAR-10 + augment | CIFAR-100 no augment | CIFAR-100 + augment |
|---|---|---|---|---|---|---|---|---|
| T(2) | 1 | Strict | - | CNN | 81.80 ±0.48 | 85.87 ±0.22 | 46.20 ±0.09 | 53.97 ±0.33 |
| | | Soft | - | **Ours** | **81.82** ±0.18 | **87.74** ±0.13 | **46.84** ±0.19 | **58.12** ±0.31 |
| SE(2) = T(2) ⋊ SO(2) | 4 | Strict | Strict | G-CNN | 81.39 ±0.26 | 85.76 ±0.21 | 44.00 ±0.18 | 49.81 ±0.56 |
| | | Strict | Partial | Romero and Lohit [24] | 83.04 ±0.25 | 84.26 ±0.55 | 46.69 ±1.02 | 52.67 ±0.40 |
| | | Strict | Soft | **Ours** | **83.21** ±0.03 | **87.02** ±0.22 | **49.47** ±0.41 | **54.48** ±0.27 |
| | 8 | Strict | Strict | G-CNN | 82.16 ±0.26 | 87.48 ±0.06 | 46.43 ±0.14 | 54.07 ±0.19 |
| | | Strict | Partial | Romero and Lohit [24] | 84.82 ±0.20 | 87.37 ±0.61 | 51.04 ±0.61 | 58.22 ±0.25 |
| | | Strict | Soft | **Ours** | **86.39** ±0.15 | **88.38** ±0.34 | **54.68** ±0.49 | **60.83** ±0.43 |
| | 16 | Strict | Strict | G-CNN | 83.33 ±0.13 | 87.15 ±0.02 | 47.68 ±0.15 | 54.20 ±0.21 |
| | | Strict | Partial | Romero and Lohit [24] | 85.92 ±0.32 | 89.48 ±0.41 | 50.58 ±0.49 | 59.63 ±0.27 |
| | | Strict | Soft | **Ours** | **86.29** ±0.29 | **89.65** ±0.22 | **54.50** ±0.26 | **60.27** ±0.31 |

# 5 Experiments

## 5.1 Verifying soft-equivariance using symmetry misspecification toy example

To verify that relaxed equivariance allows for less restrictive functions compared to strict equivariance, we repeat the MNIST6-180 problem of [24] and include our model. This toy problem is designed such that it can not be solved under strict symmetry constraints. It consists of the 6's in the MNIST dataset with half of the dataset randomly rotated by 180° degrees. The task is to determine whether a sample has been rotated, i.e., a binary classification problem between '6's and '9's. In Table 2, we compare a rotation invariant G-CNN model that consists of multiple strictly equivariant layers followed by group pooling and compare with [24] and our model. In line with our expectations, we find that in this artificial set-up the relaxed ResNet model quickly converges to 100% test accuracy. The model with strict rotation symmetry is unable to distinguish between '6's and '9's and can thus not improve over a simple coin-flip with 50% accuracy on average.

Table 2: Toy problem. Models that obey strict rotation symmetry can not distinguish between 6's and 9's whereas the task can be solved under relaxed symmetry constraints.

| SO(2) | Model | Test accuracy MNIST6-180 |
|---|---|---|
| Strict | G-CNN | 50.0 |
| Relaxed | Sparse [24] | **100.0** |
| | Soft (**ours**) | **100.0** |

## 5.2 Evaluating different fixed levels of relaxed equivariance

We assess the effectivity of relaxing equivariance constraints for translation and rotation groups T(2), SO(2) and SE(2) on CIFAR-10 and CIFAR-100 image classification tasks. As strict symmetries can lead to misspecification, we hypothesise that relaxed equivariance can improve the model performance on these tasks. As baselines, we use a strictly equivariant G-CNNs and a Partial G-CNNs [24]. The difference in model sizes arising from different kernel input dimensions are neglectable compared to the total number of model parameters for the used architecture. In Fig. 4, we plot performance on the validation set for different values of frequency parameter $\omega'$ and list final test accuracies in Table 1. From Table 1, we can see that our method outperforms the baseline on both datasets across all settings. This holds both with and without data augmentation, showing that the benefit of relaxed equivariance does not disappear when using augmentations. On T(2), we find that the improvement is negligible small and it would be interesting to see whether this gap becomes larger with increased number of basis functions or other choices of NN$_\theta$. We observe particularly large improvements in test accuracy for soft-SO(2) on CIFAR-10 and CIFAR-100. On CIFAR-100, relaxing equivariance improves test accuracy of the non-invariant baseline by 6 percentage points when using augmentation and improves by 8 percentage, without augmentation.

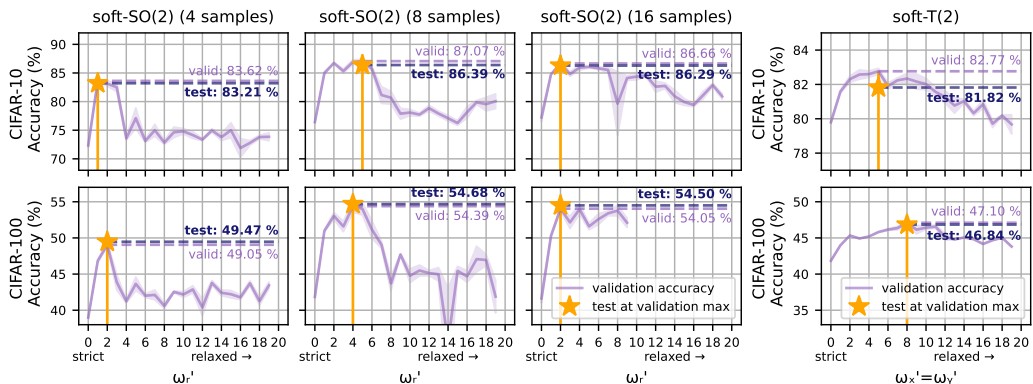

Figure 4: Relaxing equivariance constraints. Test accuracy on CIFAR-10 and CIFAR-100 is shown for $SE(2)$-equivariant models with soft-$SO(2)$ equivariances for different values of $\omega_r'$ and soft-$T(2)$ equivariance varying $\omega_x'=\omega_y'$. Some relaxation increases test accuracy in all cases, but the benefits deminish when relaxed too much. We use cross-validation to find the optimal relaxation and find that this outperforms strict equivariance constraints in all cases.

To further assess the effect of equivariance relaxation has on model performance, we can examine the impact different values for of non-stationary frequency parameter $\omega'$ on model performance. In Fig. 4, we show validation accuracy for different $\omega'$ averaged over three seeds with error bars corresponding to standard error $(\frac{\sigma}{\sqrt{3}})$ and show final test accuracy for model with best validation score. The scale of the x-axis should not directly be interpret as this depends on the chosen basis, parameterisation and initialisation of $NN_\theta$. We observe that some relaxation of equivariance with frequency parameter $\omega' \neq 0$ leads to better test accuracy compared to the strict equivariant model at $\omega'=0$. The benefits of the relaxation diminishes when equivariance is further relaxed. This is to be expected, as this then corresponds to very high frequencies in domain space which are unlikely to provide useful features anymore. We find that there exists an optimal amount of relaxation, which we can find with cross-validation. In all cases, we find that sufficiently relaxing equivariance results in higher test accuracy compared to strict equivariance. This indicates that strict equivariance might be slightly misspecified in this setting, to which our approach offers a solution.

### 5.3 Gradient-based learning of relaxed equivariance

Instead of using cross-validation, we also explore whether the right amount of equivariance can be learned using gradients. This is desirable, since cross-validation is an expensive procedure because model need to be retrained for different relaxations and requires additional hold-out validation data. Furthermore, finding the amount of equivariance using cross-validation may require sharing $\omega'$ across layers in practice to limit the search space to a single dimension, causing each layer to always have the same amount of relaxation. Training $\omega'$ with backpropagation, alongside the model parameters, is far from trivial as equivariance acts as a constraint on the functions a neural network can represent. Therefore, we can not expect that increased equivariance will lead to an improvement in terms of the regular (cross-entropy) training loss and be learned by directly optimising it with such loss. To learn equivariance constraints, which we expect will improve generalisation performance on test data, we propose to add a regularising term $\lambda\|\omega'\|_2^2$ to the objective, to encourage symmetry. The loss is inspired by the regularisation proposed in Augerino [4] to learn augmented inputs, and similarly requires an additional hyperparameter $\lambda \in \mathbb{R}$. From a Bayesian point of view, the new objective can be interpret as finding a maximum a posteriori probability (MAP) estimate after placing a Gaussian prior on the relaxation frequency parameter $\mathcal{N}(\omega' \mid \mathbf{0}, \frac{1}{2\lambda}\mathbf{I})$ (see App. C.2 for details).

In Table 3, we compare models with the partial and strict equivariance baselines, as well as the best fixed amount of equivariance found by cross-validation. We report the mean and standard error of the test accuracy on CIFAR-10 and CIFAR-100 over 3 seeds, trained with and without augmentation. For most settings of $\lambda$, the model achieves similar or improved performance over the best fixed settings found by cross-validated settings, outperforming the baselines with partial and strict equivariance. This is a promising result as it indicates that we can learn the amount of layer-wise equivariance as part of a single training procedure, without requiring expensive cross-validation or validation data.

Table 3: Learning the amount of equivariance with gradients. Comparison of soft $SE(2)$ relaxations where the amount of equivariance $\omega'$ is learned for different regularisation strengths $\lambda$. Equivariance learning achieves similar or improved test accuracy on CIFAR-10/CIFAR-100 tasks compared to the best value found by cross-validation, outperforming baselines with strict and partial equivariance.

| Group | # samples in SO(2) | T(2) | ⋊SO(2) | Model | relaxation $\omega'$ | set by | CIFAR-10 no augment | CIFAR-10 + augment | CIFAR-100 no augment | CIFAR-100 + augment |
|---|---|---|---|---|---|---|---|---|---|---|
| T(2) | 1 | Strict | - | CNN | - | - | $81.80_{\pm0.48}$ | $85.87_{\pm0.22}$ | $46.20_{\pm0.09}$ | $53.97_{\pm0.33}$ |
| | | Soft | - | **Ours** | fixed $\omega'$ | (cross-validated) | $\mathbf{81.82}_{\pm0.18}$ | $\mathbf{87.74}_{\pm0.13}$ | $\mathbf{46.84}_{\pm0.19}$ | $\mathbf{58.12}_{\pm0.31}$ |
| | | Soft | - | **Ours** | learned $\omega'$ | $\lambda$=0.001 | $79.99_{\pm0.22}$ | $87.04_{\pm0.17}$ | $46.21_{\pm0.20}$ | $55.46_{\pm0.11}$ |
| | | Soft | - | **Ours** | learned $\omega'$ | $\lambda$=0.01 | $82.04_{\pm0.16}$ | $87.28_{\pm0.17}$ | $46.45_{\pm0.15}$ | $56.32_{\pm0.35}$ |
| | | Soft | - | **Ours** | learned $\omega'$ | $\lambda$=0.1 | $82.10_{\pm0.07}$ | $87.33_{\pm0.18}$ | $46.55_{\pm0.07}$ | $56.40_{\pm0.08}$ |
| | | Soft | - | **Ours** | learned $\omega'$ | $\lambda$=0.5 | $\mathbf{82.20}_{\pm0.17}$ | $87.58_{\pm0.06}$ | $46.92_{\pm0.12}$ | $56.32_{\pm0.12}$ |
| | | Soft | - | **Ours** | learned $\omega'$ | $\lambda$=1.0 | $81.89_{\pm0.21}$ | $87.25_{\pm0.06}$ | $46.82_{\pm0.10}$ | $56.80_{\pm0.09}$ |
| | | Soft | - | **Ours** | learned $\omega'$ | $\lambda$=5.0 | $81.60_{\pm0.13}$ | $87.35_{\pm0.11}$ | $\mathbf{47.00}_{\pm0.07}$ | $57.02_{\pm0.24}$ |
| SE(2) =T(2)⋊SO(2) | 4 | Strict | Strict | G-CNN | - | - | $81.39_{\pm0.26}$ | $85.76_{\pm0.21}$ | $44.00_{\pm0.18}$ | $49.81_{\pm0.56}$ |
| | | Strict | Partial | Romero and Lohit [24] | - | - | $83.04_{\pm0.25}$ | $84.26_{\pm0.55}$ | $46.69_{\pm1.02}$ | $52.67_{\pm0.40}$ |
| | | Strict | Soft | **Ours** | fixed $\omega'$ | (cross-validated) | $83.21_{\pm0.03}$ | $87.02_{\pm0.22}$ | $\mathbf{49.47}_{\pm0.41}$ | $54.48_{\pm0.27}$ |
| | | Strict | Soft | **Ours** | learned $\omega'$ | $\lambda$=0.001 | $81.19_{\pm0.16}$ | $83.51_{\pm0.23}$ | $46.67_{\pm0.37}$ | $51.51_{\pm0.40}$ |
| | | Strict | Soft | **Ours** | learned $\omega'$ | $\lambda$=0.01 | $80.44_{\pm0.02}$ | $83.54_{\pm0.26}$ | $46.85_{\pm0.33}$ | $51.96_{\pm0.16}$ |
| | | Strict | Soft | **Ours** | learned $\omega'$ | $\lambda$=0.1 | $\mathbf{83.70}_{\pm0.16}$ | $\mathbf{86.29}_{\pm0.35}$ | $49.14_{\pm0.42}$ | $55.70_{\pm1.18}$ |
| | | Strict | Soft | **Ours** | learned $\omega'$ | $\lambda$=0.5 | $82.68_{\pm0.03}$ | $85.94_{\pm0.42}$ | $51.46_{\pm0.44}$ | $55.37_{\pm0.30}$ |
| | | Strict | Soft | **Ours** | learned $\omega'$ | $\lambda$=1.0 | $82.66_{\pm0.53}$ | $85.18_{\pm0.47}$ | $51.34_{\pm0.44}$ | $55.97_{\pm0.93}$ |
| | | Strict | Soft | **Ours** | learned $\omega'$ | $\lambda$=5.0 | $82.47_{\pm0.55}$ | $85.71_{\pm0.16}$ | $\mathbf{52.19}_{\pm0.15}$ | $\mathbf{56.34}_{\pm0.78}$ |
| | 8 | Strict | Strict | G-CNN | - | - | $82.16_{\pm0.26}$ | $87.48_{\pm0.06}$ | $46.43_{\pm0.14}$ | $54.07_{\pm0.19}$ |
| | | Strict | Partial | Romero and Lohit [24] | - | - | $84.82_{\pm0.20}$ | $87.37_{\pm0.61}$ | $51.04_{\pm0.61}$ | $58.22_{\pm0.25}$ |
| | | Strict | Soft | **Ours** | fixed $\omega'$ | (cross-validated) | $\mathbf{86.39}_{\pm0.15}$ | $88.38_{\pm0.34}$ | $54.68_{\pm0.49}$ | $60.83_{\pm0.43}$ |
| | | Strict | Soft | **Ours** | learned $\omega'$ | $\lambda$=0.001 | $83.54_{\pm0.09}$ | $86.99_{\pm0.39}$ | $50.14_{\pm0.16}$ | $56.29_{\pm0.27}$ |
| | | Strict | Soft | **Ours** | learned $\omega'$ | $\lambda$=0.01 | $83.27_{\pm0.23}$ | $87.10_{\pm0.06}$ | $50.56_{\pm0.48}$ | $56.48_{\pm0.50}$ |
| | | Strict | Soft | **Ours** | learned $\omega'$ | $\lambda$=0.1 | $86.36_{\pm0.09}$ | $\mathbf{89.26}_{\pm0.10}$ | $54.41_{\pm0.39}$ | $60.94_{\pm0.28}$ |
| | | Strict | Soft | **Ours** | learned $\omega'$ | $\lambda$=0.5 | $86.06_{\pm0.10}$ | $88.64_{\pm0.37}$ | $\mathbf{57.11}_{\pm0.23}$ | $60.67_{\pm0.29}$ |
| | | Strict | Soft | **Ours** | learned $\omega'$ | $\lambda$=1.0 | $85.71_{\pm0.34}$ | $88.90_{\pm0.06}$ | $56.28_{\pm0.11}$ | $\mathbf{60.95}_{\pm0.65}$ |
| | | Strict | Soft | **Ours** | learned $\omega'$ | $\lambda$=5.0 | $85.91_{\pm0.19}$ | $88.51_{\pm0.12}$ | $56.25_{\pm0.26}$ | $60.45_{\pm0.18}$ |
| | 16 | Strict | Strict | G-CNN | - | - | $83.33_{\pm0.13}$ | $87.15_{\pm0.02}$ | $47.68_{\pm0.15}$ | $54.20_{\pm0.21}$ |
| | | Strict | Partial | Romero and Lohit [24] | - | - | $85.92_{\pm0.32}$ | $89.48_{\pm0.41}$ | $50.58_{\pm0.49}$ | $59.63_{\pm0.27}$ |
| | | Strict | Soft | **Ours** | fixed $\omega'$ | (cross-validated) | $\mathbf{86.29}_{\pm0.29}$ | $89.65_{\pm0.22}$ | $54.50_{\pm0.26}$ | $60.27_{\pm0.31}$ |
| | | Strict | Soft | **Ours** | learned $\omega'$ | $\lambda$=0.001 | $83.39_{\pm0.12}$ | $87.87_{\pm0.27}$ | $49.24_{\pm0.55}$ | $57.22_{\pm0.18}$ |
| | | Strict | Soft | **Ours** | learned $\omega'$ | $\lambda$=0.01 | $83.59_{\pm0.09}$ | $87.96_{\pm0.19}$ | $49.63_{\pm0.48}$ | $57.87_{\pm0.25}$ |
| | | Strict | Soft | **Ours** | learned $\omega'$ | $\lambda$=0.1 | $85.81_{\pm0.16}$ | $\mathbf{89.69}_{\pm0.18}$ | $55.11_{\pm0.15}$ | $\mathbf{62.22}_{\pm0.23}$ |
| | | Strict | Soft | **Ours** | learned $\omega'$ | $\lambda$=0.5 | $85.85_{\pm0.19}$ | $89.35_{\pm0.18}$ | $55.89_{\pm0.22}$ | $61.87_{\pm0.15}$ |
| | | Strict | Soft | **Ours** | learned $\omega'$ | $\lambda$=1.0 | $85.74_{\pm0.24}$ | $89.47_{\pm0.10}$ | $55.21_{\pm0.13}$ | $61.08_{\pm0.23}$ |
| | | Strict | Soft | **Ours** | learned $\omega'$ | $\lambda$=5.0 | $\mathbf{85.95}_{\pm0.19}$ | $89.30_{\pm0.22}$ | $\mathbf{56.12}_{\pm0.18}$ | $60.87_{\pm0.18}$ |

# 6  Conclusion

In this work, we have proposed a generalisation of the group convolution that allows for a smooth parameter-efficient relaxation of otherwise strict symmetry constraints. The main idea is to use a non-stationary kernel that also depends on the absolute input group element, breaking strict equivariance constraints. Moreover, we show that we can obtain explicit control over the symmetry constraints through tunable frequency parameters, by representing group elements in a Fourier feature space.

We demonstrate relaxed equivariance in neural networks with roto-translation equivariance, relaxing the rotation and translation subgroups. We find that some relaxation of equivariance yield higher test accuracies on CIFAR-10 and CIFAR-100 image classification tasks, with and without augmentation. Furthermore, we show that the $\omega'$ parameter controlling the amount of equivariance can be learned with gradients from training data. Learning the amount of equivariance achieves similar or improved performance compared to the best value found by cross-validation and outperforms baselines with partial or strict equivariance. To perform gradient-based equivariance learning, we encourage symmetry using a similar regularisation as proposed in Augerino [4] to learn augmented inputs. Some limitations of directly regularising symmetry have been discussed in [12], such as the need for an additional hyperparameter that needs tuning and dependence on the used parameterisation of symmetry. It would be interesting to investigate whether alternative objectives [29, 33, 12] could resolve such issues. Symmetry discovery methods in literature often focus on invariances, which are easier to parameterise, whereas this work offers a way to parameterise learnable equivariance. This paves the way for automatic layer-by-layer symmetry discovery as part of a single training procedure.

By relaxing equivariance properties we can leverage useful inductive bias that symmetries can provide, while preventing possible symmetry misspecification if data does not fully obey an embedded symmetries. Experimentally, we have demonstrated that neural network layers with relaxed equivariance constraints can improve test accuracy on CIFAR-10 and CIFAR-100 image classification tasks, outperforming strict-equivariant models with up to 10 percentage points in test accuracy on CIFAR-100. We hope that the proposed non-stationary kernel as a general building block can be useful in machine learning applications that require smooth relaxations of symmetry constraints.

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
