# A Implementation

**Filter parameterisation and frequencies** All models, including the T(2)-equivariant and strict-SE(2) equivariant baselines, follow the same architecture and settings as prior work by [24]. Importantly, all models use continuous kernels that are parameterised using Fourier basis features, which is common in literature. As all baselines use the same architectures, training settings and Fourier representations for the kernels, we can attribute gains in performance solely to the relaxation of equivariance.

To select $\omega$, we use the same frequencies as used to initialise filter frequencies in [24]. To select $\omega'$ for the models with relaxed equviariance, we use cross-validation and limit the search space by considering equal frequencies along translational axes, ie. $\omega'_x = \omega'_y$, and share frequencies across layers. This scales the frequencies along the x- and y-axis equally in the Fourier basis, but does not constrain the kernel itself to be isotropic. Independent frequencies per layer is interesting future work. Lastly, to ensure the kernel is smooth in SO(2), we let $\omega_r, \omega'_r$ and associated entries in $W$ be integer valued, similar to circular harmonics. An overview of the components in $\omega$ and $\omega'$ for different groups can be found in Table 4.

**Sampling.** The integral in Eq. (2) is not tractable in closed form and we therefore choose to approximate it by using samples, similar to Finzi et al. [7]. We deterministically evaluate group elements in T(2) that correspond to (lifted) pixel locations and randomly sample 4, 8 or 16 group elements uniformly over SO(2). For [24], the uniform distribution over SO(2) covers the entire group at the start of training but has an adjustable range that is learned by early-stopping on the validation loss. All models used in the experiments use continuously parameterised kernels.

**Locally-supported in filter space.** Similar to conventional convolutional networks, we only define the kernel in a local subset $\mathcal{S}$ of the filter space, forcing the kernel to be zero $k_{\theta}=0$ outside this local support. Doing so, makes the approximation of the integral in Eq. (2) much cheaper as we only need to sample the group elements within the support. Another consequence is that we no longer obtain strictly invariant global-pooling of $Eq.$ (5). Instead, a kernel that is only a constant non-zero value in a local support represents a local-pooling layer, where the pool size equals the support size. In the experiments we let $\mathcal{S}$ be a disk in T(2) with a diameter equal to 7 pixels in the input domain $\mathcal{X}$.

Table 4: Summary of used frequency parameters for particular symmetry constraints.

| Model | | T(2) | SO(2) | $\omega$ | $\omega'$ |
|---|---|---|---|---|---|
| T(2)-CNN (regular continuous CNN) | | strict | - | $\omega_x, \omega_y$ | |
| SE(2)-CNN | | strict | strict | $\omega_x, \omega_y, \omega_r$ | |
| soft T(2)-CNN | **(ours)** | soft | - | $\omega_x, \omega_y$ | $\omega'_x, \omega'_y$ |
| soft SE(2)-CNN [soft in SO(2)] | **(ours)** | strict | soft | $\omega_x, \omega_y, \omega_r$ | $\omega'_r$ |
| soft SE(2)-CNN [soft in T(2)] | **(ours)** | soft | strict | $\omega_x, \omega_y, \omega_r$ | $\omega'_x, \omega'_y$ |
| soft SE(2)-CNN [soft in T(2) and SO(2)] | **(ours)** | soft | soft | $\omega_x, \omega_y, \omega_r$ | $\omega'_x, \omega'_y, \omega'_r$ |

# B  Training details

## B.1  Network architecture

In all experiments, we use the same architecture and training settings as prior work by [24]. The used architectures have a similar number of parameters within a 4% difference (see Table 5).

The architecture consists of a simple ResNet model [11]. A complete overview of the used architecture can be found in Sec. 5 and Fig. 3 of [24]. The network consists of a lifting layer, followed by two residual blocks with spatial max-pooling using a kernel size 2 after each group equivariant layer. After the last block, max pooling is applied over spatial and subgroup dimensions, followed by two linear layers with batch norm [13] and ReLU non-linearities.

## B.2  On kernel parameterisations

Continuous parameterisations of convolutional kernels in the context of neural networks were first proposed in [28], but require less flexible isotropic kernels. Continuously parameterised kernels with more flexible MLPs were proposed in [9], using Swish activation functions. Later works [26, 24] showed that SIRENs [30], which use a random Fourier feature basis by effectively replacing Swish activation functions with sinusoidal activations, greatly improved performance.

Random Fourier features were originally proposed in [23] to approximate (exact in the infinite-width limit) the feature basis of the radial basis function [23]. Random Fourier features are well-known and widely used within machine learning. In a Deep Learning context, they can help to overcome spectral bias and can be an effective basis when learning complex continuous signals [32, 20]. For kernel parameterisations, this encompasses kernels parameterised by small MLPs with sinusoidal activation functions in the first layer, such as SIRENs [30]. In our case, we use a shallow MLP with 32 hidden units and cosine activation functions for $NN(\cdot)$, such that the continuous kernels are the same 3-layer SIREN as used in [24].

An ablation study of different activation functions for the architecture used in this study can be found in Table 4 of Appendix F in [24]. For a more complete overview, we refer to [26] for an ablation study for regular convolutional kernels, [24] for group convolutional kernels and partial equivariant kernels, and [14] for (separable) for group convolutional kernels. We use the same Fourier feature parameterisation in all models to parameterise continuous kernels.

## B.3  Training settings

For the CIFAR-10 and CIFAR-100 datasets, we use the default train, validation and test split. We use a mini-batch size of 64 in all experiments. We optimise for 300 epochs with Adam ($\beta_1 = 0.9, \beta_2 = 0.999$)) with a learning rate of $0.001$, cosine annealed to zero with 5 epochs of linear warm-up and a weight decay of $0.0001$.

## B.4 Model parameter counts

For the strict and partial equivariant baselines, we use the same architecture as used in [24]. As can be seen from Table 5 reporting exact parameter counts, the models that were compared have approximately the same parameter counts.

Table 5: Number of parameters for different models.

| | | | | Parameter count | |
|---|---|---|---|---|---|
| Group | T(2) | $\rtimes$ SO(2) | Model | CIFAR-10 | CIFAR-100 |
| T(2) | Strict | - | CNN | 451898 | 457748 |
| | Soft | - | **Ours** | 452282 | 458132 |
| SE(2) = T(2) $\rtimes$ SO(2) | Strict | Strict | G-CNN | 451898 | 457748 |
| | Strict | Partial | Romero and Lohit [24] | 469615 | 475465 |
| | Strict | Soft | **Ours** | 469802 | 475652 |

## B.5 Runtimes

In Table 6, we report training and inference runtimes of our own implementation. Do note that these results are hardware and implementation specific. We used pytorch [21], which allows models to be efficiently run on an NVIDIA RTX 3090 24gb GPU. Adding strict rotational equivariance (SE(2) = T(2) $\rtimes$ SO(2)) is slower than only using T(2) equivariance, as it requires an extra dimension in the feature maps for samples of the rotation subgroup. Relaxing equivariance T(2) $\rtimes$ soft-SO(2), on the other hand, only requires an extra dependency in the kernel and does not come with an added computational cost. In this case, we show that relaxing equivariance can increase performance without an increase in train or inference time. We hypothesize that further engineering efforts, such as dedicated low-level CUDA implementations for (relaxed) convolutional operations, could further improve runtime performance.

Table 6: Overview of training and inference runtimes. We report the average runtime per mini-batch in seconds. Measured on CIFAR-10 on an NVIDIA RTX 3090 24 gb GPU.

| Group | No. samples in SO(2) | T(2) | $\rtimes$ SO(2) | Model | Avg. runtime Train | Inference |
|---|---|---|---|---|---|---|
| T(2) | 1 | Strict | - | CNN | 3.5 | 0.2 |
| | | Soft | - | **Ours** | 20.9 | 1.0 |
| SE(2) = T(2) $\rtimes$ SO(2) | 4 | Strict | Strict | G-CNN | 12.0 | 0.6 |
| | | Strict | Partial | Romero and Lohit [24] | 12.1 | 0.6 |
| | | Strict | Soft | **Ours** | 12.2 | 0.6 |
| | 8 | Strict | Strict | G-CNN | 29.1 | 1.3 |
| | | Strict | Partial | Romero and Lohit [24] | 29.9 | 1.4 |
| | | Strict | Soft | **Ours** | 29.4 | 1.4 |
| | 16 | Strict | Strict | G-CNN | 85.8 | 3.7 |
| | | Strict | Partial | Romero and Lohit [24] | 86.9 | 3.8 |
| | | Strict | Soft | **Ours** | 86.4 | 3.7 |

# C   Mathematical details

## C.1   Relaxed kernel inputs $(v^{-1}u, u)$ and $(v^{-1}u, v)$ equally expressive as more general $(u, v)$

To relax the convolution operator, we let the kernel not only depend on $v^{-1}u$ but on also directly on the input group element $v$ or output group element $u$: $k_1(v^{-1}u, u)$ or $k_2(v^{-1}u, v)$. We choose this form, as it allows us to parameterise the kernel in a way that can efficiently interpolate to convolutional kernels by letting it solely depend on the first argument $(v^{-1}u)$. We could consider writing our relaxed kernel in a more general form $k(u, v)$. However, upon a change-of-variables, this general form is just as expressive as the relaxed kernel in the sense that they can describe the same class of functions. To show this, we consider the reparameterisation $k = k_1 \circ f_1$ or $k = k_2 \circ f_2$. It suffices to show that bijections exist $f_1 : (a, b) \mapsto (b^{-1}a, a)$ and $f_2 : (a, b) \mapsto (b^{-1}a, b)$ between the forms:

$$(v^{-1}u, u) \overset{f_1}{\underset{f_1^{-1}}{\leftrightarrows}} (u, v) \overset{f_2}{\underset{f_2^{-1}}{\rightleftarrows}} (v^{-1}u, v)$$

If we choose inverse $f_1^{-1} : (a, b) \mapsto (b, ba^{-1})$ and $f_2^{-1} : (a, b) \mapsto (ba, b)$, it follows that

$$f_1^{-1}(v^{-1}u, u) = (u, u(v^{-1}u)^{-1}) = (u, u(u^{-1}v)) = (u, (uu^{-1})v) = (u, v)$$
$$f_2^{-1}(v^{-1}u, v) = (v(v^{-1}u), v) = ((vv^{-1})u, v) = (u, v)$$

This confirms the bijection and we conclude that kernels that depend on $(u, v)$ do, as such, not represent a broader function class than kernels that depend on $(v^{-1}u, u)$ or $(v^{-1}u, v)$. Observe that no such bijections can be found to $(v^{-1}u)$, as convolutional kernels are less expressive.

## C.2   Bayesian interpretation of symmetry regularisation.

Similar to Augerino [4], we propose to encourage symmetry through an additional regularisation term $\lambda\|\boldsymbol{\omega}'\|_2^2$ in the training objective. From a probabilistic perspective, this objective can be interpret as finding the maximum a posteriori probability (MAP) estimate after placing a Gaussian prior $\mathcal{N}(\boldsymbol{\omega}' \mid \mathbf{0}, \frac{1}{2\lambda}\boldsymbol{I})$ on the frequency parameters $\boldsymbol{\omega}'$ that control the amount of equivariance. The $\lambda \in \mathbb{R}$ hyperparameter is inversely proportional to the prior variance over relaxation parameter $\boldsymbol{\omega}'$. Note that $\boldsymbol{\omega}'=\mathbf{0}$ corresponds to strict equivariance. Hence, high values of $\lambda$ correspond to more strict equivariance whereas lower values lead to more relaxed constraints. The MAP estimate becomes:

$$\arg\max_{\boldsymbol{\theta},\boldsymbol{\omega}'} p(\boldsymbol{\theta}, \boldsymbol{\omega}' \mid \mathcal{D}) = \arg\max_{\boldsymbol{\theta},\boldsymbol{\omega}'} \left[\log p(\mathcal{D} \mid \boldsymbol{\theta}, \boldsymbol{\omega}') + \log p(\boldsymbol{\omega}')\right]$$

$$= \arg\max_{\boldsymbol{\theta},\boldsymbol{\omega}'} \left[\prod_{n=1}^{N} \log p(x_n \mid \boldsymbol{\theta}, \boldsymbol{\omega}') + \log \mathcal{N}(\boldsymbol{\omega}' \mid \mathbf{0}, \frac{1}{2\lambda}\boldsymbol{I})\right]$$

$$= \arg\max_{\boldsymbol{\theta},\boldsymbol{\omega}'} \left[\log \sum_{n=1}^{N} p(x_n \mid \boldsymbol{\theta}, \boldsymbol{\omega}') - \lambda(\boldsymbol{\omega}')^T\boldsymbol{\omega}'\right].$$

In other words, the loss that we minimise with respect to $\boldsymbol{\theta}$ and $\boldsymbol{\omega}'$ is:

$$\mathcal{L}_{\text{MAP}} = \underbrace{-\log \sum_{n=1}^{N} p(x_n \mid \boldsymbol{\theta}, \boldsymbol{\omega}')}_{\text{negative log likelihood / cross-entropy}} + \underbrace{\lambda\|\boldsymbol{\omega}'\|_2^2,}_{\text{symmetry regulariser}}$$

the sum of the cross-entropy commonly used in classification and the additional regulariser $\lambda\|\boldsymbol{\omega}'\|_2^2$. From the prior, we can intuitively see why increasing $\lambda$ encourages equivariance constraints. In the limit of infinite precision $\lambda \to \infty$, we have that the prior on the frequency components converges to a Dirac delta around zero and therefore always $\boldsymbol{\omega}'=\mathbf{0}$: the setting in which we obtain strict equivariance.