# OpenReview forum: "Relaxing Equivariance Constraints with Non-stationary Continuous Filters"
_NeurIPS.cc/2022/Conference — NeurIPS 2022 Accept_

### Official Review · Reviewer_WiTS · 2022-07-08

**Rating:** 6
**Confidence:** 3
**Soundness:** 3 good
**Presentation:** 2 fair
**Contribution:** 2 fair

**Summary:**

The paper proposes soft equivariance parameterization of neural networks that can interpolate between non-equivariant linear product thru strict-equivariant convolution to strict-invariant mapping. The contributions of the paper are twofold: 1) improving applications where strict equivariance is suboptimal; 2) providing a way to learn equivariances from data. The main idea is to use a non-stationary kernel for the group convolution. The contribution of this paper is how to parameterize that kernel. While the paper proposes a general approach, which might work for any group, in practice it only shows the idea for T(2) and SO(2). The paper obtains marginal improvements over [18].

While I like the idea, I am afraid that the paper is missing experimental and implementation details. Furthermore, can the method be extended to larger datasets and tasks?




**Questions:**

Related work: please, consider citing works on data augmentation with self-supervised training objectives as an effective way of baking in approximate equivariance into neural networks, e.g.: https://arxiv.org/abs/2111.00899.

Line 84: please, specify that $A_i$ are the basis vectors of the Lie algebra.

Line 97: please, specify that *the layers of the neural network being a group convolution* (Eq. 1) is actually the necessary and sufficient condition (following Theorem 1 in [11]). Currently, it's not clear what exactly that condition is.

Line 128: "see ??"; there is a broken link; please, fix it.

Line 161: you mean Lie group elements instead of *Lie algebra*? The exponential map maps from the group to the algebra. Actually, isn't this supposed to be a "log" map instead of "exp"?



**Limitations:**

I am not sure if the authors have addressed the limitations of the approach properly. In particular, I would like to see implementation challenges and the limitations stemming from the implementation constraints, discussed in Section 5. Ablation studies would be helpful.

**Strengths And Weaknesses:**

Pros:

1) Natural idea to interpolate between invariance, equivariance and non-equivariance.
2) Effective implementation, improving reasonable baselines.
3) Mostly well-written, with a few question (see below).

Cons:

1) Computational metrics, such as training and inference time, memory constraints and implementation details are missing.
2) The method provides only marginal gains over small datasets.

---

> ### Author Response · Authors · 2022-08-02
> **Author response**
>
> Thank you for your feedback and help in improving the paper. We are happy to answer follow-up questions.
>
> > data augmentation with self-supervised training objectives
>
> Thanks for the suggestion. We deem the reference relevant and will cite it as part of the related work section discussion on data augmentation methods.
>
> > data augmentation
>
> Data augmentation can also be used to construct invariant functions. Yet, building symmetry into neural network weights has significant advantages, which is what we try to do. To illustrate, achieving translational invariance through augmented inputs requires averaging as many forward passes as there are pixels in the input. Conversely, convnets are built by stacking layer-wise equivariant layers, and invariance can be achieved by simply pooling (over the group) in the last layer. Furthermore, layer-by-layer equivariance allows intermediate feature maps to preserve the relative pose of local features for later layers, also discussed in Cohen & Welling, 2016; Hinton et al., 2011. This is the key motivation for building equivariant layers in geometric DL literature (Bronstein et al., 2021). Typically, equivariances are fixed hard constraints that can not be adapted. We offer a relaxation that allows for explicit control over layer-by-layer equivariances.
>
> > minor questions
>
> We are thankful for spotting these unclear mistakes and statements and will fix them.
>
> Line 84: Indeed, this is the basis of the Lie algebra and will clarify this in the main text.
>
> Line 97: The condition is that the layer is equivariant. The theorem states that any (group-)convolution satisfies the equivariance constraint, but also any map that is equivariant to a (compact) group is necessarily a convolution operation (Kondor et al., 2018). We are thankful for raising this and will clearly state the condition in the main text.
>
> Line 128: we will fix the broken link.
>
> Line 161:
> Indeed, the 'log' maps from the Lie group to the Lie algebra, whereas 'exp' maps from the Lie algebra to the Lie group. To define the Lie group kernel $k : G \times G \to \mathbb{R}$ in terms of a Lie algebra kernel $\hat{k} : \mathfrak{g} \times \mathfrak{g} \to \mathbb{R}$ we need the log map since we define it as $k(g, h) = \hat{k}(\log g, \log h)$. The Lie algebra always forms a vector space. By defining a basis (Ai) in the vector space, we can describe/implement Lie algebra elements (and the associated Lie group element) in terms of real vectors.
>
> We do not say that the exponential map maps from the group to the algebra, which would be incorrect. In line 161, we say that we represent Lie algebra elements as real vectors by defining a basis in the Lie algebra. Eq. 6 shows how these real vectors of the Lie algebra relate to group elements through the exponential map, which maps from the Lie algebra to the Lie group. We choose an inverse log map from Lie group elements to unique Lie algebra elements. Consequently, we can represent Lie group elements with real vectors of the associated Lie algebra element. We will improve Sec. 4.2 by stating this more clearly. We hope this resolves the issue.
>
> > computational metrics
>
> Thanks for the suggestion. We will include memory costs and runtimes in the appendix.
>
> We show how equivariance can be relaxed through an additional dependence in the kernel. For SE(2)-equivariant networks, this dependency does not come with an additional computational cost compared to strict-equivariant neural networks, while it further improves performance. We will highlight this in the paper.
>
> > implementation details
>
> For all models, the same architecture and training settings were used in prior work Romero and Lohit. All models have an equal number of parameters within a 4% difference. For ease of reference, we will include a summary of the used architecture, training settings and other hyperparameters in the appendix.
>
> Code will be released upon acceptance.
>
> > ablation studies
>
> Regarding network design, we strived to stay as close as possible to other related works that use continuous convolutions. We refer to the CKConv paper (Romero et al., 2020) for an ablation study for regular conv kernels, Romero and Lohit, 2021 for group conv kernels and partial equivariant kernels, and Knigge et al., 2021 for (separable) for group conv kernels. Following these prior works, we use the same parameterisation of continuous kernels in the T(2)-equivariant baseline, the strict-SE(2) equivariant baseline and the proposed relaxed-equivariant model.
>
> Although these ablation studies make up most of the vital components of the used models and baselines, we understand that the references might be convoluted. We will list all relevant implementation details in the appendix alongside references to the mentioned relevant ablation studies in related work. These ablation studies should make up most relevant model components. What remains is the proposed parameter $\omega'$ controlling equivariance. We analyse this parameter in Fig 4.

---

> > ### Comment · Reviewer_WiTS · 2022-08-04
> > **Thank you for addressing my concerns**
> >
> > Thanks to the authors for taking time to answer my concerns; all of my critical questions are answered. I also read the concerns and suggestions by the other reviewers, and I agree with them. However, I think the authors need to present a sufficiently revised version of the paper, as promised.
> >
> > I think the idea and implementation would be useful to the community, thus I am raising my score.
> >
> > I would further encourage the authors to provide quantitative measurements and comparisons about the computational cost. I understand you are planning to add these in the Appendix, but I think having a rough idea of the quantitative comparisons would be helpful to the reviewing process.

---

> > > ### Author Response · Authors · 2022-08-08
> > > **Re: Thank you for addressing my concerns**
> > >
> > > We thank the reviewer for spending time reviewing our paper and are happy that the most important concerns have been resolved. We have uploaded a draft of the revised paper containing an overview of training details, quantitative measurements and other experimental details as an appendix. As promised, we will use the feedback to further polish the paper and its presentation for the camera ready.
> > >
> > > We thank the reviewer for raising the score.

---

### Official Review · Reviewer_qnnH · 2022-07-11

**Rating:** 4
**Confidence:** 3
**Soundness:** 2 fair
**Presentation:** 2 fair
**Contribution:** 2 fair

**Summary:**

The paper proposes a deep-neural network architecture for relaxing equivariance constraints. The main idea is to use a non-stationary kernel in a group convolution. They propose to model the kernels via continuous kernels parameterized via a neural network. Experiments on CIFAR10 and CIFAR100 demonstrate improvements over baseline models which are strictly equivariant.

**Questions:**

Please address the concerns in the weaknesses section above. Particularly, I am most interested in a comparison with data augmentation method, Augerino, and knowing whether the baselines use data augmentation in table 1.

**Limitations:**

The paper does not have a limitations section. It would be great to discuss some shortcomings of the approach.

**Strengths And Weaknesses:**

# Strengths
The topic of modeling parietal equivariance or learning symmetries from data is relevant and of interest to the NeurIPS community. Overall the writing of this paper is clear and the approach is clearly described. While similar to prior works in continuous kernels, the proposed non-stationary kernel is sufficiently novel.
# Weaknesses
1. The main weaknesses of this work is the empirical validation. While the proposed method is capable of modeling soft equivariances, it is unclear whether the proposed method is better than alternative approaches. For example, one could choose a model that has very few built in equivariance assumptions, e.g. transformer, and perform data augmentation. The model will learn to exhibit soft equiviarance behavior. For example, a Vision Transformer with positional encoding can be trained to be partially shift-equivariant. The current experiment lacks comparison and only conducts experiments on CIFAR10 and CIFAR 100 dataset, which is not very convincing.
2. The paper could be strengthened if additional comparisons with other baselines were made. For example, a comparison with parameter sharing method [33] or augmentation based method [A] (Augerino also has an equivariant setting (Sec. 3.1)).
[A] Benton, Gregory, et al. "Learning invariances in neural networks from training data." Advances in neural information processing systems 33 (2020): 17605-17616.
3. The authors propose to use Fourier features when parametrizing the kernel (Line. 193), it would be great to conduct an ablation study on this.
4. Some experimental and training details are missing. It might be difficult to reproduce the results. Is data augmentation being used for the baseline?
5. The authors should report the number of trainable parameters introduced and the model’s inference speed. It seems like the proposed approach would be slower than a standard ResNet.

# Miscellaneous
* The checklist should have been attached with the main paper and not in the supplemental materials.
* Some references still cites the arXiv version instead of the conference version. For example, Deep-Sets [32] is published at NeursIPS.
* Line 18. "euclidean" --> Euclidean
* Line 75. "continuosly" --> continuously
* Line 186. "fourier" --> "Fourier"

---

> ### Author Response · Authors · 2022-08-02
> **Author response**
>
> > 1a building in assumptions
>
> This is a very interesting point.
>
> Models with very few built-in equivariance assumptions, such as Transformers, have recently shown very impressive results on complex tasks when trained on enough data. This raises the question: to what extent are strong inductive biases like strict equivariance constraints required, and could higher performance be achieved when such assumptions are dropped? In this context, it seems natural to develop architectures that can interpolate such modes. Our method enables precisely this: a method to explicitly control the amount of equivariance in a conv layer.
>
> Transformer models have recently outperformed convnets on large vision benchmarks, but counter-examples do exist (e.g. Zhuang Liu et al., 2020). As a research community, we should keep developing multiple approaches. Our contribution has the potential to fundamentally change the underlying symmetry properties and behaviour of a successful class of DL models.
>
> > 1b invariance using data augmentation
>
> Invariant functions can also be constructed by augmenting the input and aggregating results. However, this is inefficient as intermediate layers do not respect symmetry constraints. To illustrate, achieving translational invariance this way requires averaging as many forward passes as pixels in the input. Feature maps in convnets are equivariant and invariance can be achieved by simply pooling (over the group) in the last layer. Furthermore, layer-by-layer equivariance allows intermediate feature maps to preserve the relative pose of local features for later layers, further discussed in Cohen & Welling, 2016; Hinton et al., 2011. This is the key motivation for building equivariant layers in geometric DL literature (Bronstein et al., 2021). Typically, equivariances are fixed hard constraints that can not be adapted. We offer a relaxation that allows for explicit control over layer-by-layer equivariances.
>
> > 2 Augerino
>
> Augerino (Benton, 2020) focuses on learning invariances from data. It relies on data augmentation, which, as discussed, can be an inefficient representation. Our contribution allows relaxed symmetry constraints to be incorporated into individual layers. Our parameterization may be combined with an Augerino-like objective or act as building block in other invariance learning methods to parameterise differentiable equivariance constraints. This is interesting future work.
>
> In Sec 3.1 of Benton et al., 2020, a method is discussed to extend Augerino to equivariant outputs. This is useful in segmentation tasks where the final output transforms equivariant w.r.t. the input. However, it does not pose an efficient way to embed symmetry in intermediary layers and cannot be applied to classification problems. Therefore, we do not include this as a baseline in our experiments.
>
> > 3 parameterising kernels with Fourier features
>
> Our main contributions are the general relaxation of the conv operator Eq. 2, demonstrating interesting special cases (Eqs. 3,4,5), and showing that we can explicitly control the symmetry with $w'$ (Sec. 4.2).
>
> The use of a Fourier feature basis for kernels, as such, is not a contribution. In fact, this is a common way to parameterise continuous kernels for regular group equivariance in literature (Romero et al., 2020, Knigge et al., 2021). From a DL perspective, this encompasses kernels parameterised by small MLPs with sinusoidal activation functions in the first layer, such as SIRENs. Compared to Romero & Lohit, our approach is theoretically more expressive and achieves higher performance.
>
> Ablation studies for kernel parameterisations with different activation functions can be found in the literature. We refer to the CKConv paper (Romero et al., 2020) for an ablation study for regular conv kernels, Romero and Lohit for group conv kernels and partial equivariant kernels, and Knigge et al., 2021 for (separable) for group conv kernels. We use the same Fourier feature parameterisation in all models.
>
> > 4 training details and reproducibility
>
> For all models, the same architecture and training settings were used as prior work Romero & Lohit, 2021. For ease of reference, we will include a summary of the used architecture, training setting and hyperparameters in the appendix.
>
> Code will be released upon acceptance.
>
> > 5 number of parameters and training speed
>
> Thank you for the suggestion. All models have an equal number of parameters within a 4% difference. We will add parameter counts and runtimes to the paper.
>
> Strict-SE(2) equivariant and relaxed soft-SE(2) equivariant are slower than the normal T(2)-equivariant baselines if regular representations are used but typically achieve higher performance. For the SE(2) equivariant model, the additional dependency does not come with an additional computational cost. Here relaxed equivariance improves performance without increasing train and test time, which is a benefit of our approach. We will highlight this more properly.

---

> > ### Author Response · Authors · 2022-08-08
> > **Author response**
> >
> > In addition to addressing the concerns above, we have uploaded a revised paper using feedback from reviewers. The version fixes the most important issues and contains an appendix with additional details.
> >
> > Previously our baselines were run without data augmentation. Inspired by comments of qnnH, the revised version Appendix C contains the same experiments but with data augmentation. From the runs completed so far (some cells still missing), we conclude that our relaxed equivariant models still perform best. This shows that the benefit of relaxed equivariance does not disappear when using data augmentation.
> >
> > Thank you again for your feedback and help in improving the paper. We would appreciate a response to our rebuttal so that we still have time to address follow-up questions or concerns you might have.

---

> > > ### Comment · Reviewer_qnnH · 2022-08-08
> > > **Thanks for the response**
> > >
> > > Thanks for the response and partially address my concerns. I remain unconvinced on the following:
> > > 1. A baseline with param-sharing approach [39, was 33] on more significant dataset would be beneficial. This was not directly response by the authors.
> > > 2. I disagree the authors are Augerino shouldn't be compared. Specifically, in Augerino's introduction states
> > > > Augerino (2) can discover partial symmetries.
> > >
> > > Also, Augerino can be applied to classification tasks.
> > >
> > > 3. The additional reported results in Appendix C indicate that augmentation improves both the baseline and proposed method. Can the authors elaborate on why data augmentation also benefits the proposed method?
> > >
> > > Due to these concerns, I am maintaining my original evaluation.

---

> > > > ### Author Response · Authors · 2022-08-08
> > > > **Re: Thanks for the response**
> > > >
> > > > We thank the reviewer for engaging in the rebuttal discussion. We are glad that the earlier response has partially addressed concerns. We are happy to answer follow-up questions on any remaining issues.
> > > >
> > > > > 1
> > > >
> > > > Thank you for pointing this out. Parameter-sharing approaches rely on representing weights in a fully-connected manner. This does not scale to large neural networks as the memory cost would be equivalent to representing each convolutional layer as a fully-connected layer. Also, it requires knowing the sizes of input feature maps
> > > >
> > > > Our work offers a way to parameterise relaxed equivariance constraints efficiently in a way that allows explicit control over the amount of equivariance. We will make sure this fundamental benefit of our method over parameter-sharing methods is clear in the main text.
> > > >
> > > > > 2
> > > >
> > > > The purpose of Augerino is _to adjust_ invariance/equivariance. The point of our paper is to find an efficient layer-wise parameterisation of invariance/equivariance. So the papers are looking at different things. An invariance learning method like Augerino can benefit from new parameterisations of equivariances, like our approach where equivariance constraints can be adjusted for individual layers. However, this is orthogonal to our contribution.
> > > >
> > > > To clarify our earlier statement: Augerino focuses on learning partial invariances through data augmentations. The _way_ that these invariances are parameterised is through data augmentation. This approach can be applied to image classification but does not consider equivariance in intermediate layers, which we focus on. There does exist an 'equivariant setting' of Augerino which considers transformed outputs, but as explained in the rebuttal, this version can not be used for image classification.
> > > >
> > > > > 3
> > > >
> > > > The data augmentation results show that data augmentation and relaxed equivariance incorporate inductive biases that are different to a certain extent. The fact that our method still outperforms equivariant models even with data augmentation shows that our parameterisation helps take full advantage of the inductive bias implied by data augmentation.

---

### Official Review · Reviewer_bGts · 2022-07-11

**Rating:** 5
**Confidence:** 4
**Soundness:** 3 good
**Presentation:** 2 fair
**Contribution:** 3 good

**Summary:**

Strict equivariance constraints in deep learning may sometimes be overly restrictive for some tasks. This paper proposes a relaxation of equivariance by allowing the convolutional kernel to be non-stationary (thus depending on the absolute input location). They achieve so by introducing frequency parameters, which can interpolate between strict equivariance and element-wise linear product, thus adjusting the "softness" of equivariance. They empirically demonstrate that their proposed soft equivariance leads to better performance on CIFAR10 and CIFAR100 in terms of classification accuracy.

**Questions:**

1. How much performance gain comes from the usage of random Fourier features rather than the soft equivariance? Can the authors consider running an ablation study for this?
2. I am very confused by the concept of "kernel" in this paper. On the one hand, it is only the convolutional filters, so it does not need to satisfy kernel properties such as positive semi-definite (See https://people.eecs.berkeley.edu/~jordan/kernels/0521813972c03_p47-84.pdf). However, on the other hand, the authors mention GPs and lengthscales, and use the random Fourier features. All of these are for the "kernels" in kernel methods literature. Furthermore, even though they are using random Fourier features, the neural networks take in the concatenation of features and directly produce a scalar as output. In this case, the kernel function is merely an arbitrary function on a pair of inputs.

I am ready to raise my score if the authors could clarify my confusion and give a convincing justification.

**Limitations:**

I have no concern over negative societal impact.

**Strengths And Weaknesses:**

**Strength**
1. This paper studies an important problem: strict equivariance constraints can be overly restrictive and is misspecified for some problems. It would be nice to have soft equivariance and automatically adjust symmetries on data. This paper provides a simple, practical and parameter-effective solution.
2. This paper shows clear performance improvements using the proposed soft-equivariance on CIFAR10 and CIFAR100. It would have a great impact if this improvement is widely applicable to lots of tasks.
3. The proposed solution is new and it seems to solve the over-strict equivariance problems. (However, I think more careful ablation study and potentially more experiments on new datasets are needed in the future)

**Weaknesses**
1. Although this paper shows clear performance improvements for the proposed soft equivariance compared to strict equivariance on CIFAR10 and CIFAR100, it is not convincing enough that the performance gain actually comes from soft equivariance. It would be ideal to have an ablation study showing that, the performance gain does not come from using the random Fourier features. It would be more convincing if the authors could show performance improvements on more tasks and datasets. But I understand that this is not possible during the rebuttal period. So I am not asking the authors to run experiments on extra datasets, just a suggestion for future versions.
2. This paper is not careful enough with mathematical terminology: For example,
    * (line 166) "cross-product of two vector spaces". I believe they mean the product space of these two spaces, rather than the vector product here (https://en.wikipedia.org/wiki/Cross_product).
    * (line 175) "let $\hat{k}$ be a Gaussian Processes (GPs)". Firstly, $\hat{k}$ is a kernel function while GP is a stochastic process with a mean function and kernel function. It does not make sense to me by saying a kernel function is a stochastic process. Secondly, the kernel function here corresponds to the convolutional filter, which is different from the kernel functions in GP literature. In GPs, the kernel function needs to be positive semi-definite. Thirdly, I think the authors only use the kernel function here and it has nothing to do with GPs (the probabilistic model)
    * (eq 3) I believe it is only an element-wise product. For me, it is a bit contrived to say this corresponds to the fully-connected layers. Even if you apply the averaging pooling layers, you still have only one output unit?

(Please correct me if I misunderstand)

---

> ### Author Response · Authors · 2022-08-02
> **Author response**
>
> Thank you for your feedback and help in improving the paper. We are happy to answer follow-up questions.
>
> > gain in performance because of relaxation
>
> We compare our soft-equivariant approach with strict-equivariant neural networks. All models, including the T(2)-equivariant and strict-SE(2) equivariant baselines, follow the same architecture and settings as prior work by Romero and Lohit (2021). Importantly, all models use continuous kernels that are parameterised using Fourier basis features, which is common in literature. As all baselines use such Fourier representations, we can attribute the gains in performance solely to the relaxation of equivariance.
>
> We realise that we did not clearly state that baselines also use Fourier features to represent continuous kernels. We are sorry for any confusion caused and understand how this might have affected the initial assessment. We will fix this by stating this more clearly in the paper.
>
> > on the use of Fourier features
>
> Our main contributions are the general relaxation of the convolution operator Eq. 2, showing that this relaxes equivariance with linear products, equivariance and invariance as special cases (Equations 3, 4, 5) and a parameterisation that allows explicit control through w' (Sec. 4.2).
>
> The use of random Fourier feature basis for kernels, as such, is not a contribution of ours. In fact, this is one of the most common ways to parameterise continuous kernels in for regular group equivariance in literature (Romero et al., 2020, Knigge et al., 2021). From a deep learning perspective, this encompasses kernels parameterised by small MLPs with sinusoidal activation functions in the first layer, such as SIRENs. We offer an efficient generalisation of continuous (group) convolutions that can be explicitly controlled. Compared to Romero and Lohit, our approach is theoretically more expressive and achieves higher performance.
>
> Ablation studies for kernel parameterisations with different activation functions can be found in the literature. We refer to the CKConv paper (Romero et al., 2020) for an ablation study for regular convolutional kernels, Romero and Lohit, 2021 for group convolutional kernels and partial equivariant kernels, and Knigge et al., 2021 for (separable) for group convolutional kernels. We use the same Fourier feature parameterisation of continuous kernels in all models.
>
> > typos and unclarities
>
> Thank you for spotting these. Typos and mistakes have been fixed.
>
> > relation to Gaussian Processes (GPs)
>
> These are really helpful points. Perhaps we emphasised the inspiration for our parameterisation too much in the discussion. We will tailor the presentation more to what the method actually does.
>
> Convolutional kernels are very different from kernels used in GPs. Indeed, GP kernels need to be positive semi-definite (PSD), whereas convolutional kernels have no such constraint. Unfortunately, the same word 'kernel' is used in literature for both the convolutional kernel and the PSD GP kernels, whereas they refer to two distinct mathematical objects.
>
> Although the kernel of a GP has to satisfy PSD constraints, samples taken from the process do not have such constraints. Our parameterisation was inspired by thinking of placing a GP prior (with a particular PSD kernel) on the convolutional filter. In our paper, 'be a GP' should have been 'place a GP prior on $\hat{k}$', which has likely contributed to the confusion. Another issue with our language is that it implies Bayesian inference, which we do not do. In the end, we simply perform MAP implied by the same GP kernel (Rahimi et al. 2017 show that particular PSD kernels can be approximated with finite random Fourier basis functions). This has confused the presentation a bit.
>
> Although this is an interesting interpretation, it is not vital to our contribution. To improve the presentation, we will therefore remove references to GPs. We will rewrite this form of a perspective where filters are implicitly regularised by parameterising them in terms of RFF basis functions (implied by a particular PSD kernel).
>
> > relation to a fully-connected layer
>
> We thank the reviewer for raising this important point. An element-wise product followed by average pooling would be equivalent to performing a dot product, optionally with added bias. As rightfully pointed out by the reviewer, this only results in a single scalar output. However, if we do this over a collection of channels, the output channels form not a scalar but a vector of which each element is the result of a different dot product, optionally with added bias. This is akin to the output features of a fully-connected layer where the output is a vector and elements correspond to the result of a dot-product with a row vector in the weight matrix, optionally with added bias. We will add that given that if we have multiple channels and regard them as the output features, it behaves the same as a fully-connected layer. We hope this resolves the issue.

---

> > ### Comment · Reviewer_bGts · 2022-08-05
> > **Response to Author Rebuttal**
> >
> > Thanks for spending time answering my questions. My main technical concern has been addressed by the authors, and I have therefore raised my score to 5. I am mostly convinced by the experimental results and I think the proposed method has a positive contribution to the community and demonstrates interesting results. However, the manuscript needs significant work before publishing. The authors need to reconsider their way of presenting and the use of terminology & notations. I hope the authors are willing to spend significant time polishing their writings.

---

> > > ### Author Response · Authors · 2022-08-08
> > > **Re: Response to Author Rebuttal**
> > >
> > > We thank the reviewer for spending time reviewing the paper. We are glad that most technical concerns could be addressed in the rebuttal. We uploaded a draft of the revised paper, which should address the most critical concerns. As promised, we will use the feedback to further polish the presentation for the camera ready.
> > >
> > > We thank the reviewer for raising the score.

---

### Official Review · Reviewer_hLit · 2022-07-15

**Rating:** 7
**Confidence:** 4
**Soundness:** 4 excellent
**Presentation:** 3 good
**Contribution:** 3 good

**Summary:**

This paper proposes a parameter-efficient neural layer for relaxed equivariance. The main idea is to use a non-stationary integral operator whose kernel depends on absolute group elements as well as relative group elements.  This allows to interpolate between non-equivariant, strictly-equivariant, and strictly-invarint mappings in learning. The proposed method is validated on CIPAR-10 and CIFAR-100 image classification benchmarks, outperforming other strictly and partially equivariant methods.


**Questions:**

I think this is a good paper, but addressing the weaknesses I described above would strengthen the contribution.
Please respond to my points of the weaknesses.


**Strengths And Weaknesses:**

Strengths
- Clear motivation for relaxed equivariance
- Simple yet effective idea of non-stationary filter
- Enabling targeted relaxation of a specific subgroup
- Convincing results on MNIST and CIFAR image classification

Weaknesses
- Missing comparison to related work. The proposed method is compared only with G-CNN and Romero and Lohit [18] while there exist other recent related work, e.g., E(2)-CNN, Implicit Equivariance in Convolutional Networks.
- Limited experimental setup. The experiments are done using a small network on small datasets. It is unclear whether the performance gain is retained when bigger networks (ResNet18, 50) and larger datasets (ImageNet) are used. Furthermore, the benefit of equivariance relaxation is not demonstrated beyond image classification.
- Missing analyses. Comparison of FLOPs and # of parameters and ablation studies on design choices (activation function..) are not provided.
- How to set non-frequency parameter w? While the advantage of the proposed method is the controllability of equivarience levels, the performance of a learned model looks significantly affected by the control parameter w as shown in Fig. 4. The sensitivity of performance to w and the determination of w in practice need to be discussed.

---

> ### Author Response · Authors · 2022-08-02
> **Author response**
>
> Thank you for your feedback and help in improving the paper. We address your concerns below and are happy to answer follow-up questions or discuss further.
>
> > missing Implicit Equivariance in Convolutional Networks
>
> We thank the reviewers for pointing out this work. We deem the paper relevant and will add it to the related work section. The approach could also be interpreted as a type of approximate equivariance but, in contrast to our work, requires an additional equivariance error loss in the objective function. The method was not proposed to improve modelling data that does not fully respect group symmetry but rather to incorporate symmetries that are more difficult to parameterise. Since the method has another purpose, and it is unclear how the extra loss should be balanced or controlled for our use case, we do not include it for comparison.
>
> > limited experimental setup
>
> To our knowledge, no relaxations of equivariance with steerable kernels, such as in E2-CNN, have been proposed. We, therefore, only compare to Romero & Lohit (2021) as this is the only other method to incorporate relaxed equivariance into neural network layers for deep neural networks. We show theoretical and practical improvement over this method. We agree with the reviewer that it would be exciting to scale to larger datasets, such as ImageNet. Doing this will require some more engineering, but can use the parameterisation we suggest. We do think that the paper already has a concrete message that is demonstrated, i.e. that we can relax equivariance constraints in a controllable manner and in a way that is more efficient than previous approaches (e.g. Romero and Lohit, 2021).
>
> > missing analysis
>
> In all experiments, we use the same architecture and training settings as prior work by Romero and Lohit, 2021. All models have an equal number of parameters within a 4% difference. For ease of reference, we will add a complete list of used training settings, hyperparameters, runtimes, and parameter counts in the appendix.
>
> Code will be released upon acceptance.
>
> > computational complexity
>
> We add runtimes to the appendix. Adding strict rotational equivariance (SE(2) = T(2) ⋊ SO(2)) is slower than only T(2) equivariance as it requires an extra dimension in the feature maps for samples of the rotation subgroup. Relaxing equivariance (T(2) ⋊ soft-SO(2)), on the other hand, only requires an extra dependency in the kernel, which does not come with an additional computational cost. In this case, we show that relaxing equivariance can increase performance without an increase in train or inference time.
>
> > $\omega'$' parameter
>
> Our method shows how equivariance constraints can be relaxed in a way that can be explicitly controlled by a $\omega'$ hyper-parameter. We tune $\omega'$ using cross-validation, just like how many other hyperparameters need to be learned in deep learning. In Fig 4, we assess the impact on model performance for different values of $\omega'$. We find that model performance is sensitive to the amount of equivariance, as expected. In all our experiments, some relaxation results in higher test performance than strict equivariant models. We will further elaborate on this in the main text.

---

### Meta-Review · Area_Chair_vAAd · 2022-08-30

**Recommendation:** Accept
**Confidence:** Less certain

**Metareview:**

This paper received initially quite mixed reviews, but after a strong rebuttal from the author's, a number of reviewers increased their scores, leaving it with an overall borderline positive rating. The work was praised for the interesting and novel core idea, the potential significance of the work's contribution, generally clear writing, and providing good empirical results for small-scale image datasets. Concerns that remained after the author's rebuttal included the lack of experiments on larger scale datasets or architectures, issues in the presentation/care of some mathematical formulations and notation, and a lack of comparison to certain potential benchmark approaches.

Taking these into account, my own view of the work is that the strengths outweigh the negatives. In particular, I feel like most of the remaining concerns raised are either not reasonable or not appropriate grounds for rejection. In particular, I do not think that direct comparisons to Augerino are warranted beyond the new augmentation experiments in Appendix C (which I would consider promoting to the main paper for the final version) or that the issues with the mathematical formulations are severe enough to be reasonable grounds for rejection. Given that the paper has some clear strengths that all reviewers agree on, my recommendation is therefore that the paper should be accepted.

**Award:**

No

---

### Decision · Program_Chairs · 2022-09-14

Accept